# Bacterial Endophyte Community Dynamics in Apple (*Malus domestica* Borkh.) Germplasm and Their Evaluation for Scab Management Strategies

**DOI:** 10.3390/jof7110923

**Published:** 2021-10-31

**Authors:** Shahid A. Padder, Sheikh Mansoor, Sajad A. Bhat, Tawseef Rehman Baba, Rauoof Ahmad Rather, Saima M. Wani, Simona Mariana Popescu, Shakeela Sofi, Malik Asif Aziz, Daniel Ingo Hefft, Othman M. Alzahrani, Ahmed Noureldeen, Hadeer Darwish

**Affiliations:** 1Division of Basic Sciences and Humanities, FoH, Sher-e—Kashmir University of Agricultural Sciences & Technology of Kashmir, Srinagar 190025, Jammu and Kashmir, India; drsajadbhat@skuastkashmir.ac.in (S.A.B.); Drsaimams22@gmail.com (S.M.W.); Shakeelasofi1221@gmail.com (S.S.); 2Division of Biochemistry, FBSc, SKUAST-J, Jammu 180009, Jammu and Kashmir, India; 3Division of Fruit Science, SKUAST-Kashmir, Srinagar 190025, Jammu and Kashmir, India; tawseefrehman27@gmail.com; 4Division of Environmental Sciences, FoH, Sher-e—Kashmir University of Agricultural Sciences & Technology of Kashmir, Srinagar 190025, Jammu and Kashmir, India; rauoofahmad@gmail.com; 5Department of Biology and Environmental Engineering, University of Craiova, 13, A.I. Cuza, 200585 Craiova, Romania; popescu_simona83@yahoo.com; 6Division of Basic Sciences and Humanities FoA, Sher-e—Kashmir University of Agricultural Sciences & Technology of Kashmir, Wadura Sopore 193201, Jammu and Kashmir, India; malikazizaa21@gmail.com; 7University Centre Reaseheath, Reaseheath College, Nantwich CW5 6DF, UK; daniel.hefft@reaseheath.ac.uk; 8Department of Biology, College of Science, Taif University, P.O. Box 110099, Taif 21944, Saudi Arabia; o.alzahrani@tu.edu.sa (O.M.A.); a.noureldeen@tu.edu.sa (A.N.); 9Department of Biotechnology, College of Sciences, Taif University, P.O. Box 110099, Taif 21944, Saudi Arabia; hadeer@tu.edu.sa

**Keywords:** scab, apple, endophytes, *Venturia inaequalis*, germplasm

## Abstract

The large genetic evolution due to the sexual reproduction-mediated gene assortments and propensities has made *Venturia inaequalis* (causing apple scab) unique with respect to its management strategies. The resistance in apple germplasm against the scab, being controlled for by more than fifteen genes, has limited gene alteration-based investigations. Therefore, a biological approach of bacterial endophyte community dynamics was envisioned across the apple germplasm in context to the fungistatic behavior against *V. inaequalis*. A total of 155 colonies of bacterial endophytes were isolated from various plant parts of the apple, comprising 19 varieties, and after screening for antifungal behavior followed by morphological, ARDRA, and sequence analysis, a total of 71 isolates were selected for this study. The alpha diversity indices were seen to fluctuate greatly among the isolation samples in context to microflora with antifungal behavior. As all the isolates were screened for the presence of various metabolites and some relevant genes that directly or indirectly influence the fungistatic behavior of the isolated microflora, a huge variation among the isolated microflora was observed. The outstanding isolates showing highest percentage growth inhibition of *V. inaequalis* were exploited to raise a bio-formulation, which was tested against the scab prevalence in eight apple varieties under controlled growth conditions. The formulation at all the concentrations caused considerable reductions in both the disease severity and disease incidence in all the tested apple varieties. *Red Delicious* being most important cultivar of the northwestern Himalayas was further investigated for its biochemical behavior in formulation and the investigation revealed different levels of enzyme production, chlorophyll, and sugars against the non-inoculated control.

## 1. Introduction

Apple (*Malus domestica Borkh*) belongs to the plant family *Rosaceae*, which is comprised of more than 100 genera and 300 species across the world, most frequently established in temperate climates [1]. All across the globe, apples are concentrated in temperate and subtropical countries, with trivial production levels in hilly areas of tropical parts of the globe. The area in Jammu and Kashmir utilizes 136.54 thousand hectares for its apple production. Annual production reaches 1672.72 thousand metric tons and productivity of 12.50 tons per hectare [2,3].

Several fungal pathogens such as *Venturia inequalis*, *Podosphaera leucotrichia*, *Alternaria mali*, etc., cause a huge loss to the apple industry. Of these, *V. inaequalis* (Cooke) G. Wint causing apple scab is economically the most significant pathogen. It is known to cause economic injury levels in many apple-producing countries across the globe. In the Jammu and Kashmir state, this disease was reported in 1970s and ever since, it has caused havoc in the apple industry with the extent of fruit loss of up to 60%, estimating an amount in hundreds of crores [4]. *V. inequalis*, being a heterothallic haploid ascomycete, reproduces both sexually and asexually [3,4,5,6]. The disease takes the severe form in regions in which recurrent rainfall through the spring seasons leads into ascospore emancipation and infection [5]. In spring, the predisposing factors such as temperature and moisture conditions are favorable and ascospores are both released and disseminated by wind to bring about epidemics. When an ascospore lands on a susceptible leaf or fruit, it form lesions, leading to the formation of mitospores (conidia), which are splashed by water or blown by wind to bring the secondary infection. The ascospore landing is marked by the switch from pathogen to saprophytic to the parasitic phase [7,8]. Many scab control strategies across the globe are aimed at protecting plants against primary infections through chemical-mediated management means [9]. As the disease is polycyclic in nature, multiple fungicide applications are needed to assess this disease [10]. It shall be noted that some limited research studies have managed to contain the disease by spraying only five chemical-based fungicidal compounds at critical growth stages of the crop [4,5]. Systemic fungicide sprays limited to seven in number have previously been able to contain the disease but due to the emergence of fungicide resistance in pathogens, these practices have become obsolete [6]. A recent survey in southeastern Europe has revealed the development of resistance in *V*. *inaequalis* against the potent fungicides dodine, pyraclostrobin, boscalid, difenoconazole, and cypradonil [11].

The modern horticultural practices demand chemical-centered strategies for phytopathogen management and emphasize alternative control measures focused on the use of products that induce resistance against the diseases [12], as the present pesticide intensive strategies have led to ecological degradation, inducing detrimental environmental effects beyond the huge effect on the economy of the farming community. One of the alternatives to these practices could be natural chemicals based on copper or sulfur, but due to their efficacy on pathogens as well as the advent of phytotoxicity apprehensions, these compounds have also had little impact on scab management strategies. One of the windows of management could be provided by the use of resistant cultivars but such cultivars are observed to be less popular due to their poor yield and consumer acceptance. Additionally, *V. inaequalis* possess huge genetic evolution on account of the sexual reproduction-mediated gene hodge-podges, which has resulted in the breakdown of major scab resistance gene Vf among its populations [13]. Despite huge spending on research centered on the biological control of phyto-pathogens across the world and regardless of well-founded publications on this topic, biological control remedies for apple scab are not in practice, except for a few limited investigations, e.g., the use of *Cladosporium cladosporioides* H39 to contain the scab in canopy [14] and priming mediated through fructans. Some other scanty studies on biological remedies of scabs have been conducted but these investigations have targeted the primary infection causing ascospores (overwintering) in litter and not the actively multiplying disease on canopy [15,16]. While apple endophytes have been used widely against apple scab in Germany, Belgium, the Netherlands, and Canada [17], the non-availability of convenient germplasm in these areas has limited such studies; in addition, these investigations have tested the fungal endophytes, which could face a serious challenge in the context of the changing climate scenario of the Jammu and Kashmir state. The endophytes in general are interesting entities for understanding the plant microbe interfaces that augment plant development and protect it from phytopathogens [18].

In contemplating global climate change and the elevated temperatures in the present investigated areas, the bacterial microflora has a larger role to play in future biological remedies for phytopathogens. Endophytes are more competent than their rhizospheric counterparts due to their huge diversity in well-established anabolic pathways, which has a direct bearing on its ability to be used as a successful tool in phytopathogen management strategies. Thus, the objective of this investigation was to visualize the bacterial endophyte community dynamics across the apple germplasm in the context of fungistatic behavior against *V. inaequalis*, as well as to test the hypothesis that potential antagonists as a formulation at different concentrations may affect the scab in the tree canopy of apple. The gene cascades that indirectly modulate the containment of phytopathogens of the host have also been studied in the current investigation.

## 2. Materials and Methods

### 2.1. Sampling Area

The four districts of the Kashmir valley (Jammu and Kashmir state, India), namely Anantnag (3984 km^2^ area, 33.730N 75.150E, 1615.44 m elevation), Baramulla (4588 km^2^ area, 34.20N 74.340E, 1592.885 m elevation), Pulwama (1090 km^2^ area, 33.87160N 74.89460E, 1651.711 m elevation), and Shopian (612 km^2^ area, 33.720N 74.830E, 2057.095 m elevation) were surveyed. The area falls under temperate climate with an average annual temperature of 11.8 °C and 2853 mm of rainfall.

### 2.2. Sampling Methodology

In order to isolate *V. inaequalis*, fully expanded apple leaves (counting from the 5th, 6th, and 7th tip) were collected from different apple varieties with scab symptoms of nearly black spots with a brown center, irregular margins on the upper surface, and subtle lesions with discrete margins on the lower surface were selected as per the protocol [7]. Isolation of the pathogen was carried out as previously reported by Machardy and Mansoor et al., [7,16]. The collected infected leaves were incubated at 21 °C with wet paper towels under sterile conditions until sporulation appeared. Then, these sporulated leaves were cut into pieces of the size of approximately 5 mm and were surface sterilized with 70% ethanol for 1 min and 10% sodium hypochlorite for 5 min, followed by three washings in sterile distilled water. The sample pieces were dried and placed on potato dextrose agar media, after which the plates were incubated at 21 °C and mycelium was transferred on fresh potato dextrose agar (PDA) plates a checked for purity. The conidia were observed under microscope to identify *V. inaequalis* as per the protocol explained by Sutton et al., [19]. The cultures were grown on potato dextrose agar (PDA) for 14 days for single spore isolation. The geminated spore containing petri plates were subjected to incubation at 21 °C until sporulation and conidia were observed as described above. All the platings were done in triplicates.

### 2.3. Pathogenicity on Detached Leaves

*V. inaequalis* was grown on PDA plates for 14 days at 21 °C. The conidia were washed from the plates using sterilized distilled water and the spore count of the order of 4 × 10^5^/mL was adjusted using a hemocytometer. Counting from the 5th, 6th, and 7th tip, fully expanded leaves were collected from three-year-old different apple varieties (Red Delicious, Gala Redlum, Early Red One, Red Fuji, Gala Mast, Red Chief, and Summer Red) on the upper surface. A 100 µL of the conidial suspension was placed on the scratched portion, the leaves were incubated for 14 days at 21 °C, and symptoms of the scab development were observed [16] Conidia were washed from the leaves and observed under microscope for their resemblance to *V. inaequalis*. The experiment was carried out in triplicates.

Fresh roots, twigs, and leaves of different apple varieties were collected in order to isolate bacterial endophytes. In order to reduce the chances of contamination, the plant materials looking afresh (free from any disease) were used for endophytic bacterial microfloral isolation.

### 2.4. Isolation, Morphological Characterization, and Preservation of Bacterial Endophytes

The surface sterilization technique [20] with some minor modifications was employed for isolating the bacterial endophytes. The samples (roots, twigs, and leaves) after surface disinfection were placed for grinding with sterile mortar and pestle, and then the ground samples were dissolved in phosphate buffer saline (1000 mL containing 1.44 g Na_2_HPO_4_, 2.4 g KH_2_HPO_4_, 0.2 g KCl, and 8.0 g NaCl, and with a pH of 7.4). Ground plant material was then serially diluted and a volume of 100 µl from apiece dilution of 10-2, 10-3, 10-6, and 10-9 was spread plated on solid growth media (trypticase soy agar, nutrient agar, and luria bertani agar), followed by an incubation at 37 °C for a period of 24–72 h. The experiment had three replications.

### 2.5. V. inaequalis Inhibition upon Co-Cultivation with Bacterial Endophytes

The full-grown bacterial endophytes (48 h old) on the nutrient agar Petri dishes were placed (as line streaks) on PDA at equal distances of 2.2 cm from the center of the 100 × 15 mm Petri dish. A 5 mm disc taken from the 10-day-old *V. inaequalis* culture was placed at the center of the Petri dish as per the protocol of [21] with some modifications and the plates were incubated at 23 °C. The plates without the bacterial endophytes were kept as the control. The experiment was repeated with three replications set for the experimental design.

The percent growth inhibition was calculated using the formula Gv − (Gv + endo)/Gv × 100, where Gv is the daily growth rate of *V. inaequalis* alone over six days and Gv + endo is the daily growth rate of *V. inaequalis* cultivated together with an endophyte over the period of one week, as per the protocol of [22] with slight modifications. All the isolates with growth inhibition activity against *V. inaequalis* were selected for further study.

### 2.6. Diversity Quantification and Molecular Characterization with 16S rRNA and ARDRA Analysis

Genomic DNA of the isolated bacterial endophytes was extracted by growing single bacterial colonies on their particular medium of isolation for 24 h at 37 °C and then the procedure of DNA extraction [23] with some modifications was employed. For PCR amplification, a total volume of 5–10 µL of the aqueous phase (upper) was used as a source of the DNA template [23]. The remaining mixture was stored at −20 °C. DNA amplification was performed for the 16S rRNA region using the primers 27F, ‘AGAGTTTGATCCTGGCTCAG-3′ and 1492R, and 5′-GGTTACCTTGTTAGGACTT-3 [24].

Restriction fragment length polymorphism relying on the restriction-mediated analysis of amplified 16S rRNA coding DNA, i.e., amplified 16S ribosomal DNA restriction analysis (ARDRA), was employed to distinguish the isolates prior to sequencing. The restriction digestion of amplified 16S ribosomal DNA with RsaI and HhaI (Thermo Scientific) was conducted using the method of [25] with minor adjustments. All the procedures were performed in triplicates. The unique strains as evident from the ARDRA analysis were selected for sequencing. The nucleotide sequences were compared with the sequences from the NCBI database and sequences displaying >99% homology were retrieved by the Nucleotide-Basic-Local-Alignment-Search-Tool (BLAST N) package, which is accessible through the server of the National Center for Biotechnology Information (NCBI) (www.ncbi.nlm.nih.gov/BLAST) (accessed on 7 August 2018). For the construction of the Neighbor Joining Tree, all the 16S rDNA sequences obtained were trimmed, applying trim primers for mapped reads (only primer track and reads that hit the primer parameters were selected) using CLC software (QUAGEN Bioinformatics). Multiple cluster alignment and phylogenetic analysis were performed using MEGA software (v.10.2.6) based on the neighbor joining method using a 1000 repetition bootstrap. The diversity indices (alpha diversity) of the isolated taxonomic groups were calculated using EstimateS (v. 9.1.0). The phylogenetic analysis on distance tree results and graphics of the outstanding sequences were carried out against the GenBank database using the NCBI BLAST tool available at http://blast.ncbi.nlm.gov/Blast.cgi (accessed on 7 August 2018).

### 2.7. Plant Growth Promoting Activities

#### 2.7.1. In Vitro 1-Aminocyclopropane-1-carboxylic Acid (ACC) Deaminase Activity of Bacterial Endophytes

ACC deaminase activity of the isolated bacterial endophytic strains was determined by employing the method of Penrose and Glick [26]. ACC deaminase activity was determined through the colorimetric method and screening of the microflora was also done for the presence of the ACC deaminase gene (acdS). Both experiments were performed in triplicates.

#### 2.7.2. Detection of acdS (ACC Deaminase Gene)

In order to amplify the ACC deaminase gene, the precise primers ACCDF, 5′-ATGTCACTGTTGGAAAAGTTCGA-3′ and ACCDR, and 5′-TCAGCCGTCC CTGTAATAGC-3′ were designed. The genomic DNA of the isolated strains were used as a template in the reaction process of PCR. The temperature sets used for PCR amplification were: 95 °C for 5 min for the initial denaturation; 30 cycles of the denaturation; at 94 °C for 60 s; annealing at 53 °C for 90 s; and extension at 72 °C for 10 min. The sequence of the obtained amplicons was checked against the GenBank database using the NCBI BLAST tool available at http://blast.ncbi.nlm.gov/Blast.cgi (accessed on 24 July 2019).

#### 2.7.3. Biological Nitrogen Fixation Assay

In order to identify nitrogen fixing bacteria, they were grown on minimal nitrogen-free media [27]. Two methodologies were used for the purpose of bacterial growth in nitrogen-free media (BNF) and PCR-based identification of the nifH gene (component II of nitrogenase enzyme). The bacterial strains were screened for the presence of the nifH gene and was carried out as follows: the degenerate primers nifH F2 (5′-CAGAACACCATYATGGARATGG-3′) and nifH R2 (5′-CGCCGAGSACGTC TAGAAG-3′) were designed from available bacterial nifH gene sequences observed online, and amplification was done for some selected endophytic bacterial strains. The sequence of the amplicons were analyzed against the GenBank database using the NCBI BLAST tool available at http://blast.ncbi.nlm.gov/Blast.cgi (accessed on 3 December 2019).

Within the method of bacterial growth in nitrogen-free media, the endophytic strains were observed by growing on minimal nitrogen-free media [27,28]. The growth of a particular strain on minimal nitrogen-free media specified the presence of a nitrogen-fixing attribute.

#### 2.7.4. Phosphate Solubilization

All isolates were grown in tryptic soy (TS) broth. Log-phase growing cells (O.D. 0.6) of each culture (15 µL) were spotted on Pikovskaya’s medium plates [29] and were incubated in triplicates at 28 °C for 3–4 days. The zone of solubilization was measured and colony size was also measured to calculate solubilization index by the formula: SI = colony diameter + halo-zone diameter/colony diameter [30]. To carry out the assay for phosphate estimation, the method given by [28] was employed with three replications. The soluble phosphorus formed was estimated calorimetrically.

#### 2.7.5. Indole-3-acetic Acid (IAA) Production

IAA production was assessed by Salkowski’s method with minor modifications in the incubation period (2 days instead of 4 days) [31]. The standard curve ranging from 5 µg/mL to 300 µg/mL was used for calculating IAA production.

#### 2.7.6. Production of Ammonia and Hydrocyanic Acid

For the detection of ammonia, the method of Demutskaya and Kalinichenko [32] with slight modifications (incubation of 72 h) was employed. The freshly grown culture of bacterial strains (O.D. 0.6) were inoculated in 10 mL peptone water broth in triplicates and were incubated for 72 h at 30 °C. A volume of 0.5 mL of Nessler’s reagent was added and the appearance of a yellow/brown color indicated ammonia production.

For hydrocyanic acid (HCN) detection, the method of Schippers [33] was adopted using Kings B medium supplemented with 0.44% of glycine. The experiment was carried out in triplicates. HCN production was detected with the transformation in color of the filter paper (plated above the streaked strains), which was previously dipped in 2% of Na_2_CO_3_ prepared in 0.05% of picric acid; the color change was observed visually from yellow to dark brown.

#### 2.7.7. Siderophore Estimation Using the Chrome-azurol-S (CAS) Liquid Assay Method

For the detection of siderophores, the Chrome-azurol-S (CAS) method was employed. In this method, 0.1 mL of bacterial cell-free extract of the supernatant (obtained by inoculating 15 μL of 3 × 10^9^ CFU/mL log-phase cells (O.D. 0.6) into nutrient broth and after incubation (48 h) was centrifuged for 15 min at 10,000 rpm) was mixed with Chrome-azurol-S (CAS) solution (0.5 mL), along with the shuttle solution (10 µL of 0.2 M 5-sulfosalicylic acid), in triplicates. The contents were kept at room temperature (for 10 min) and the change in color (to yellow) of the solution indicated the existence of siderophore production.

#### 2.7.8. Formulation Development

For the formulation development, the top ten isolates based on their ability to arrest the growth of *V. inaequalis* were selected to develop the formulation. The compatibility of the isolates was checked against each other and then these endophytic bacterial isolates were characterized as per the procedures in Bergey’s manual of determinative bacteriology [34]. Formulation was achieved by taking the log-phase bacterial cells and centrifuging at 12,000× *g* for 15 min; the pellets were washed with autoclaved distilled water to remove the media components and finally the pellets were resuspended in sterile distilled water with concentrations of 0.5, 1.0, 1.5, 2.0, 2.5, 3.0, 3.5, 4.0, 4.5, 5.0, 10, and 15%. The suspension of pellets was diluted by 10-folds before making the final concentrations in order to include the carrier materials to be tested in the future with the formulation.

### 2.8. In Vivo Studies

#### Effect of Isolated Cultures on the Disease Incidence and Severity on Leaves under Pot House Conditions

The plants of various varieties (three years old) of apple were procured from the Division of Fruit Science SKUAST-K, Shalimar Srinagar, and Lidder valley nurseries. These plants were grown in pot culture/controlled conditions and for this experiment, soil was autoclaved at 121 °C and with 15 psi for 30 min. The plant roots were treated with the selected bacterial strains by dipping the seedlings in separate concentrations of bacterial suspensions for half an hour. The control plants were dipped for the same time in sterile distilled water only. The foliar applications of all the concentrations were made by using a Honda Ltd. mechanical sprayer (droplet size of 325 microns) as per the recommended schedule of sprays for scab management. Spore suspension of the *V. inaequalis* with a density of 10^6^ spores/mL was applied (30 days interval) as droplets on the whole plants in order to ensure the pathogen presence in the environment of the tested plants. Sterile distilled water was sprayed in the same way on the control plants instead of on the formulation.

The disease incidence and severity were recorded every three weeks by counting the number of infected leaves on young shoots and rosettes, and visually estimating their percentage diseased area. Observations were made for five randomized branches per plant and ten older leaves were evaluated for leaf scab [35] to estimate the disease incidence and severity using the following formula: % disease incidence = number of scabbed leaves/total number of tested leaves × 100.

The disease severity (S) depended upon the modified scale using the Townsend-Heberger’s formula, expressed as
S (%) = Σ(nivi)/NV × 100
where n is the degree of infection according to the following scale: 0 = no attack, 1 = 1–3 spots per leaf, and 2 = more than three spots per leaf.

The experiment was carried out in a completely randomized design with five replications (five plants each treatment).

### 2.9. Biochemical Alterations Post Endophyte Inoculation

One month after inoculating the endophytic formulation, a total of eight leaves (at the third internode from the top of the plant) were randomly selected for the quantification of physiological and biochemical attributes (namely guaiacol peroxidase (GPOD), catalase (CAT), superoxide dismutase (SOD), ascorbic acid peroxidase (POD), esterase (EST), and acid phosphatase (ACP)) of the Red Delicious variety of apple. The method of Liu et al., [36] was employed and described fleetingly as follows: roughly 0.15 g (fresh leaf tissue) was homogenized in a pre-cooled mortar in 5 mL of phosphate buffer (50 mmol/L, pH 7.8) solution. The homogenate obtained was centrifuged at 11,000.00× *g* for a period of 15 min at 5 °C. The resulting supernatant was used to calculate the enzyme activity. For the POD assay, one unit (enzyme) was defined as the content of the enzyme, resulting in a 1% absorbance surge in 1 min at 470 nm. CAT activity was determined by measuring the rate change of H_2_O_2_ absorbance in 1 min (at 240 nm). For the SOD assay, one unit (enzyme) activity amounted to the content of the enzyme that resulted in the 50% inhibition rate of nitroblue tetreazolium reduction. The ascorbic acid peroxidase (POD) activity was calculated as per the method of Nakano and Asada [37], and enzyme activity was stated as A290/min/g fresh mass. Esterase activity was estimated spectrophotometrically at 25 °C using 2-napthylacetate as per the protocol of Radic and Pevalek-Kozlina [38]. It was calculated by observing the elevation in absorbance at 313 nm due to the formation of 2-napthol. The reaction mixture contained 1.0 mL of 100 mM Tris (pH 7.4) and 15 µL of 100 mM 2-napthylacetate, and both were dissolved in methanol (absolute). For each measurement, 30 µL of crude extract was used. The enzyme activity was expressed as µmol of hydrolyzed substrate per minute/mg protein. Acid phosphatase activity was measured by the procedure of Besford [39] and absorbance was measured at 420 nm as units/min/g fresh weight. The chlorophyll estimation was done on the same extract as discussed above and was measured as previously described by Walker [38]. The sugars, namely glucose and fructose, in the leaves were determined calorimetrically at 340 nm absorbance as per the method of Kunst-Wilson and Zajonc [40].

### 2.10. Data Analysis

All the experiments were conducted in a completely randomized design (CRD) and the experimental data are expressed as mean ± SD from five separate observations/replications. Normality and homogeneity of the collected data was tested using Levene’s Test in IBM SPSS Statistics 19.0 (SPSS, Inc., Chicago, IL, USA). Differences in the culturable fungistatic microflora against the *V. inaequalis* was assessed using the one-way analysis of variance technique. The principal component analysis (PCA) was estimated using XLSTAT (V 2021.1). The post hoc analysis of the data was carried out by Tukey’s test.

## 3. Results

### 3.1. Identification and Diversity of Isolated Endophytic Microflora

A total of 155 colonies based on various colony characteristics, namely elevation, margin, color, and texture, were isolated from various plant parts of the apple plant, comprising 19 varieties (Appendix A). After screening for antifungal behavior, followed by morphological, ARDRA, and sequence analysis, a total of 71 isolates pooled from all the sources belonging to 23 genera (Figure 1; Appendix A) were selected for this study. The genera shared certain qualities among the different sampled parts, namely leaves, twigs, and roots, as presented in Figure 2. The phylogenetic tree based on the Neighbor Joining method of 16S rDNA gene sequences of isolated microflora is expressed in Figure 3. A total of 59.15% of the bacteria were observed to be gram-negative and 40.85% were gram-positive, thus the population isolated was dominated by gram-negative microflora. The Simpson diversity index was 0.8529, 0.8933, and 0.9216 in the sampled twigs, roots, and leaves, respectively. Similarly, the Shannon index was observed to be 1.855, 1.877, and 1.984 in the sampled twigs, roots, and leaves, respectively, and the Margalef index was 2.94, 2.551, and 3.806 in the sampled twigs, roots, and leaves, respectively. All the indices were measured with respect to fungistatic microflora against V. inaequalis. The alpha diversity indices are presented in Figure 4.

### 3.2. Population Dynamics among the Collected Samples

The population of bacterial endophytes in twigs varied from Log_10_ 4.70 × 10^5^ to Log_10_ 7.21 × 10^5^ (cfu/gFW), with the highest observed in Fenna and the lowest in Red Delicious. In the roots, the population varied from Log_10_ 5.84 × 10^5^ to Log_10_ 8.27 × 10^5^ (cfu/gFW), with the highest observed in Maharaji and the lowest in Summer Red. In the leaves, the population varied from Log_10_ 5.35 × 10^5^ to Log_10_ 7.83 × 10^5^ (cfu/gFW), with the highest observed in Maharaji and the lowest in Golden Delicious (Appendix A).

### 3.3. Antagonist Activity against V. inaequalis

Screening of the isolates was conducted based on antifungal activity against V.inaequalis. Among all the isolates across the available apple plant germplasm, namely Red Delicious, Golden Delicious, Red Chief, Gala Redlum, Fuji Zehn Aztech, Maharaji, Red velox, Gala mast, Malus floribunda, Summer Red, Red Fuji, Priscella, Oregon Spur, Gavin, Lal Cider, Manchurian, Fenna, and Ambri, it was observed that a total of 71 isolated strains exhibited antifungal behavior expressed as a percentage of growth inhibition (PGI) against *V. inaequalis*, with the highest PGI exhibited by the isolate *P. psychrophila* strain DST28 (96.35 ± 0.03%), followed by the *P. fragi* strain DST29 (91.25 ± 0.11%), while the lowest PGI was exhibited by the isolate *O. pectoris* strain DST59 (11.25 ± 0.08%) (Table 1). In the screening process, we found that the highest number of fungistatic bacterial strains were isolated from Golden Delicious, followed by Fenna, and the least were found from the variety Fuji Zehn Aztech and Gala Mast (Appendix A).

### 3.4. Assay for In Vitro Plant Growth Promotion of Indole-3-acetic Acid Production

All the isolated bacterial strains were screened for their ability to produce auxin, i.e., IAA (Table 1). The culture supernatant of 32 isolates produced IAA. The IAA production varied from 1.16 ± 0.23 (µg/mL) to 146.05 ± 0.53 (µg/mL). The highest IAA production was observed in the B. aryabhattai strain DST scab (146.05 ± 0.53 µg/mL) and the lowest production was observed in the isolate *Variovorax boronicumulans* strain DST18 (1.16 ± 0.23 (µg/mL).

### 3.5. ACC Deaminase Production

In the genetic screening for the acdS (ACC deaminase synthase) gene, a total of 19 bacterial species were found to be positive, showing sequence similarity to the acdS gene of other bacterial species. An endophytic bacterial strain, specifically the P. fluorescens strain smppsap5, was taken as a positive control. In the plate assay, the B. subtilis strain DST50 was observed to exhibit the highest ACC deaminase activity (22.61 ± 0.72 α-ketobutyrate µM/g/h), followed by the B. amyloliquefaciens strain DST scab4 (19.31 ± 0.72 α-ketobutyrate µM/g/h), and the lowest activity was exhibited by the Povalibacter uvarum strain DST24 (6.41 ± 1.02 α-ketobutyrate µM/g/h) (Table 1).

### 3.6. Biological Nitrogen Fixation

All the isolates were screened for the nifH gene and four strains, namely the B. subtilis strain DST10, Klebsiella pneumoniae strain DST36, B. velezensis strain DST42, and Micrococcus yunnanensis strain DST70, were found to be positive, showing sequence similarity to the nifH gene of other bacterial species. A total of 13 isolates, namely the B. velezensis strain DST scab7, B. aryabhattai strain DST Scab9, B. subtilis strain DST10, P. veronii strain DST21, B. foraminis strain DST26, P. psychrophila strain DST28, P. fragi strain DST29, Paenibacillus lautus strain DST31, B. subterraneus strain DST33, K. pneumoniae strain DST36, B. velezensis strain DST42, P. libanensis strain DST64, and M. yunnanensis strain DST70), were positive for nitrogen fixation.

### 3.7. Phosphate Solubilization

All the isolated bacterial endophytes were screened for phosphate solubilization ability and it was observed that only 14 isolates released phosphate from tri-calcium phosphate (Table 1). Additionally, it was observed that quantitative solubilization varied from 102. 16 ± 1.05 (µg/mL) to 225. 51 ± 1.01 (µg/mL) in the O. anthropi strain DST61 and P. fluorescens strain DST22, respectively, while the solubilization index varied from 1. 34 ± 0.18 in the P.libanensis strain DST64 to 2. 53 ± 0.01 in the Pseudomonas fragi strain DST29, and was statistically at par with the P. veronii strain DST21 and P. peoriae strain DST32 with the solubilization indexes of 2. 52 ± 0.09 and 2. 52 ± 0.19, respectively.

### 3.8. Qualitative Estimation of Siderophore Production, Ammonia Secretion, and Hydrogen Cyanide Production

All the isolated bacterial endophytic strains were screened for NH_3_ production, N_2_ fixation, siderophore production, and HCN production, and it was observed that among all the isolated strains, a total of 20, 13, 16, and 11 isolates were observed to be positive for NH_3_ production, N_2_ fixation, siderophore production, and HCN production traits, respectively. A multivariate analysis (principal component analysis (PCA)) was used as a tool to gain insight into the complexity of the bacterial microflora so as to analyze the outputs of each. The results of the qualitative screening indices were converted to binary codes (1, isolate positive to the test; 0, isolate negative to the test) to run the PCA (Table 2). The number of microbial groups was confirmed by the cluster analysis approach and this resulted in the formation of 12 clusters (Figure 5) based on similarity indices.

The output of such an analysis (cluster and PCA analysis) contains two parts: strain distribution in the factorial space and the table of all the isolates included in different groups (A to L). Figure 6 displays variable and case distribution for isolated microflora; the analysis accounted for 57.621% of the total variance. The first factor (with 1.303 eigenvalue) was positively related to NH_3_ production and siderophore production (loadings of 0.523 and 0.805, respectively), while the second factor (with 1.028 eigenvalue) was related to N_2_ fixation and HCN production, with positive (0.717) and negative loadings (0.00–0.717). A total of 12 phenotypic groups were formed by the isolated microflora on the basis of the qualitative traits. The isolates in group C were all positive to the qualitative traits (*Bacillus* sp.) and group E (33 strains) included bacterial isolates negative to all the traits, and these groups clustered in the bi-plot as per the homogeneity with respect to the qualitative traits. The PCA (Figure 6) suggested a high level of biodiversity and complexity within the isolated microflora in the context of the traits discussed. Thus, this provides a clear representation regarding which traits in the microflora move in a similar direction and hence offers insights into the complexity of the screening process of microflora (most of the fungistatic strains were negative to these qualitative traits).

### 3.9. Effect of Isolated Cultures on the Disease Incidence and Severity on Leaves under Pot House Conditions

The top isolates showing the highest percentage growth inhibition of *V. inaequalis*, which included the *B. aryabhattai* strain DST scab, *B. amyloliquefaciens* strain DST scab4, *B. velezensis* strain DST scab7, *P. xylanexedens* strain DST14, *Rahnella aquatilis* strain DST15, *P. psychrophila* strain DST28, *P. fragi* strain DST29, *B. subterraneus* strain DST33, K. pneumoniae strain DST36, and *Leclercia adecarboxylata* strain DST39, were selected for formulation development. The isolates were found to be compatible with each other when grown as a consortium and these isolates had huge variation with respect to different molecular and biochemical attributes (Table 3 and Table 4). Then, these isolates were grown together to make the formulation. The formulation at all the concentrations caused considerable reductions in both the disease severity and disease incidence in all the tested apple varieties, which included Red Delicious, Gala Redlum, Early Red One, Red Fuji, Gala Mast, Red Chief, and Summer Red. In Red Delicious, the disease intensity was lowest (44.21 ± 0.12) at 15% formulation concentration and was statistically similar (45.09 ± 0.13) with 10% formulation concentration against the control (98.19 ± 0.05). In Gala Redlum, the disease intensity was lowest (32.12 ± 0.21) at 15% formulation in comparison to the control (91.95 ± 0.15) and some degree of homogeneity was observed across the apple germplasm tested with minor contradictions (Figure 7). Similarly, the developed endophytic formulation had the significant impact of disease severity. In Red Delicious, the disease severity was lowest (17.28 ± 0.24) at 15% formulation concentration, which was statistically at par (17.33 ± 0.16) with the 10% formulation concentration against the control (77.56 ± 0.12). A similar trend appeared for all apple varieties tested with some infinitesimal unconventionalities (Figure 8).

### 3.10. Biochemical Alterations Post Endophyte Inoculation

Investigated cultivar Red Delicious, being the most important cultivar of the Kashmir valley, revealed different levels of enzyme production, chlorophyll, and sugars upon inoculation with the bacterial formulation against the non-inoculated control (Figure 9. The peroxidase (GPOD) content was observed to be 8.82 (A470/min g FW) in 15% formulation concentration against the control [18.19 (A470/min g FW)]. Furthermore, superoxide dismutase (SOD), catalase (CAT), acid phosphatase (ACP), ascorbic acid peroxidase (POD), and Esterase (EST) content varied largely upon inoculation with the formulation (Figure 10). Glucose content was observed to be highest (774.09 µg/g FW) at 15% formulation concentration and lowest in the untreated control (349 µg/g FW). A similar trend was seen in the fructose content, while the chlorophyll content was observed to be highest (2.31 mg/g FW) in the plants treated with the 15% formulation concentration and lowest in the untreated control (1.27 mg/g FW), and chlorophyll b varied in a similar fashion. Correlation studies on these parameters revealed that there was a significant correlation observed with different magnitudes of coefficients, as represented by the correlation matrix (Pearson) and by coefficients of determination (Figure 10).

In order to have a better understanding of the relationship between the formulation concentration and the biochemical response, which was not identified by the correlation analysis, and to determine the influence of each variable, principal component analysis (PCA) was applied (Figure 11). The eigenvalue of principal component 1 and principal component 2 were greater than 1 (Figure 11a) and thus were considered for the PCA. The PCA of GPOD, CAT, SOD, POD, EST, ACP, DS1, chlorophyll, glucose, and fructose with respect to different formulation concentrations (C) in Red Delicious revealed that principal component 1 (F1) and principal component 2 (F2) accounted for 97.08% and 1.68% of the data variance, respectively, as shown in biplot and Bootstrap ellipses (Figure 11). The first factor was related to acid phosphatase (represented by ACP in the biplot), ascorbic acid peroxidase (POD), glucose (Glu), fructose (Fru), chlorophyll a (Chl.A), disease severity (DS1), and Chlorophyll b (Chl.B), with loadings of 0.718, 0.720, 0.608, 0.657, 0.710, 0.752, and 0.719, respectively. The second factor was related to peroxidase (GPOD), superoxide dismutase (SOD), catalase (CAT), and esterase (EST), with loadings of 0.674, 0.587, 0.623, and 0.522, respectively. The principal component scores on PC1 had the same sign between the disease severity and formulation concentrations up to the formulation concentration magnitude of 3% and then changed to opposite signs beyond this concentration, depicting the prominent effect of formulation concentration beyond this level. The overlapping scores of the last two concentrations (concentration of 10% and 15%) revealed the peak effect values of formulation (Figure 11b, bootstrap ellipses). As there was huge variation in the data due to heterogeneity in the measurement scales, the data was normalized by Z-score normalization.

## 4. Discussion

*V. inaequalis*, which causes apple scab, is economically the most important pathogen in apple-based farming systems worldwide. Scabs are considered to be the most important disease in apple in terms of yield loss, which can reach as much as 70% [16]. The chemical-based management strategies currently employed across the globe have raised public concerns over pesticide residue in foods, thereby generated a great deal of interest in reducing their use. The present management strategies have resulted into ecological degradation and economically burdens farming communities. Therefore, some economically realistic and biological-by-origin strategies need to be developed for the proficient management of this disease to thereby reduce the health hazards of pesticides on the consumer. Bacterial endophytes are one of the most significant non-pathogenic entities concomitant with plants across the family lines. Besides being the prominent candidate of endosymbiosiome, bacterial endophytes have the potential to attract the most interest among almost all the biological phytopathogen management strategies in changing the global climate scenario. These aspects have made this strategy one of the low cost and ecologically sound tools in the integrated crop disease clampdown. As they are native candidates to plant niches, they are known for not disturbing the existing microfloral equilibrium. They are also extremely versatile in metabolism, which has made them the most interesting entity to researchers across the world. Therefore, their antifungal behavior as well as the presence of various genes and metabolites as a central mechanism to their mode of action have been investigated in this research study. A total of 150 strains isolated from various apple varieties were studied in this investigation, collected across the apple germplasm centers of Jammu and Kashmir, India. However, we limited our study to only 71 isolates based on their capability to inhibit the growth of *V. inaequalis* in vitro, expressed as a percentage growth inhibition that varied from 96.35 ± 0.03% to 11.25 ± 0.08%. The inhibitory activity of bacterial endophytes against *V. inaequalis* in the present study was much more advanced than as previously reported [14,15]. In the screening process, we found that the highest number of fungistatic bacterial strains were isolated from Golden Delicious germplasm, followed by Fenna, and lowest number derived from the variety Fuji Zehn Aztech and Gala Mast, thus giving an indication to the existence of heterogeneity among the sampled cultivars in the context of scab resistance. The extent of endophyte colonization in plants by endophytes was observed to be correlated to disease resistance [33]. Liu et al., [41] reported that the endophyte community-shaping in apple is being determined by the type of tissue, the sampling site, and cultivar, which in turn have a prominent impact on the number of potential antifungal biota strains present. The plant endosymbiosiomes have been widely used to isolate microflora with biocontrol ability [42] and the studies on such entities have revealed their abilities to contribute in conferring resistance to crops against various phytopathogens, including other substantial benefits on the host such as nutrient acquisition, enhanced growth, and improved resistance to heat, drought, and other stresses [43]. The present study revealed that there is an abundant diversity of bacterial microflora across the Himalayan apple germplasm with respect to their antifungal behavior against *V. inaequalis*. Among the isolated fungistatic microflora, the endophytes of the family *Gracilicutes* were found to be abundant in numbers, totaling to 45.15% of the total isolates. Based on the 16S rRNA gene sequences of the isolated bacterial endophytes prominently belonging to the *Bacillus*, *Pseudomonas*, *Enterobacter*, and *Paenibacillus* genus, the endophytic lifestyle of these genus have been widely reported in many crops [44,45]. The apple plant has been widely observed to niche the endophytes that have huge potential to contain the pathogenic fungal growth and their relative proportion is dependent on the type of cultivar and plant part selected for sampling [46]. Although the present investigation revealed the denser proportion of fungistatic bacterial microbiota in comparison to previous reports, which could be attributed to the more diverse apple germplasm of the Himalayan foot hills of India that were selected for the isolation of the investigated bacterial endophyte strains, among the pooled isolated microbiota, a total of 59.15% of the bacteria were observed to be gram-negative and 40.85% to be gram-positive. Zhang et al., [47] reported the predominance of gram-negative bacteria among the plant tissues, although some researchers reported contrary results to our findings [48]. The bacterial microflora in different scab-infested plant parts resulted in the changed genetic and physiological behavior; therefore, the characteristics of diversity and the physiology of the pooled fungistatic microbiota could help us to understand their ability to ameliorate the success of the host against diseases. In the present investigation, all the collected microflora were visualized and in this context, the diversity indices, namely the Simpson index, Shannon index, Margalef index, and other indices, varied greatly. Studies on the putative hyper-diversity indices regarding endophytes in all major linages of terrestrial plants have revealed fluctuating results on all the major diversity indices [49], wherein the abundance, diversity, richness, and communities may largely be influenced by the habitat of the plantations [50], thereby validating our findings on the investigated facet of diversity. The population of bacterial endophytes in the sampled tissues varied from Log_10_ 4.70 × 10^5^ to Log_10_ 8.27 × 10^5^ (cfu/gFW), which agrees with the population ranges in previous studies of endophytic bacteria in many crops [51,52]. The variation in the bacterial endophytic populations experienced robust fluctuations that prevalently depended upon factors including the type of germplasm screened, the physiological transitions, the dynamics among the host plants, and the soil environment attributes such as nutrient dynamics, organic matter fluctuations, plant genetic constitution, tissue type, and plant ecology [53].

All the isolates were subjected to the determination of their plant growth-promoting abilities and it was found that the culture supernatant of 32 isolates produced IAA, which varied from 1.16 ± 0.23 (µg/mL) to 146.05 ± 0.53 (µg/mL). Our findings are in agreement with many researchers [54,55]. In our study, an impressive number of strains (19) were positive to the existence of the *acdS* gene and a majority among them were Firmicutes but only nine were seen to produce ACC deaminase in liquid assay; one of the possible reasons for this could concern the absence of some important inducers that are needed for *acdS* expression. Among the nine strains, the *B. subtilis* strain DST50 exhibited the highest ACC deaminase activity (22.61 ± 0.72 µM/g/h) and the *Povalibacter uvarum* strain DST24 (6.4 ± 1.02 µM/g/h) exhibited the lowest activity in this context. The activity reported in the current investigation is considerably higher than previously reported *B. subtilis* strains [56], which were only 448.3 ± 2.91 nM α-ketobutyrate/mg/min. In a similar fashion, all the isolates were screened for the nitrogen-fixing attribute and a total of 13 isolates were observed to be positive for nitrogen fixation, but the sceening for the presence of the *nifH* gene revealed that only four strains (*B. subtilis* strain DST10, *K. pneumoniae* strain DST36, *B. velezensis* strain DST42, and *M. yunnanensis* strain DST70) were found to be positive, showing sequence similarity of the *nifH* gene to other bacterial species [57]. The detection in the other strains could not be made even after using degenerate primers from the known *nifH* gene and this could be possibly attributed to the high divergence in *nifH* gene sequences. The thirteen strains were seen to utilize the atmosphere-embedded nitrogen and remarkably, seven of them were isolated from the shoot (leaves and twigs). Some bacterial endophytic microflora from the stem and leaves have been investigated for effective nitrogen fixation [58,59] and interestingly, these shoot-associated endophytic microflora had the advantage over their root bacteria counterparts as they inhabited the less competition-based aerial niche for nitrogen fixation and growth [59].

In the present investigation, the *P. fluorescens* strain DST22 showed maximum phosphate solubilizing ability, which was 225. 51 ± 1.01 (µg P/mL), and the solubilization index was highest in the *P. fragi* strain DST29 (2.53 ± 0.01). Although many strains of *P. fluorescens* have been observed to solubilize phosphate [60], in the present study, *P. fluorescens* showed far higher phosphate solubilization and a far higher solubilization index than earlier reported strains such as the *P. fluorescens* strain PMS1 [61]. Bacterial endophytes play a pivotal role in plant development in controlling the phosphorus availabilities [62]. The release of copious acids (organic) such as gluconic acid results in the acidification of soils and causes phosphate solubilization from various sources in the soil solution. Gluconic acid has been found to convert inorganic phosphate into ortho-phosphorus acid, which is the available form to be taken up by the plants. In the qualitative detection of some important plant growth-promoting traits among the isolated bacterial endophytes, a total of 16 strains produced siderophores and ammonia was produced by a total of 20 isolates, while hydrogen cyanide was produced by a total of 11 isolates. It was also reported that there is siderophore production among the bacterial isolates in a similar range [62]. Other researchers have also widely documented the siderophore production process of bacterial endophytes [63]. In the present investigation, the studied bacterial endophytic strains could prevent phytopathogens from obtaining an adequate quantity of iron due to their ability to produce siderophores, therefore limiting their capability to thrive and multiply. This mechanism of induced iron starvation can be the possible reason for their antifungal behavior. The iron chelation through this mechanism does not impact the plant development as plants grow at significantly poorer iron echelons than the invading pathogenic microflora [64]. Even in certain cases, plants can take up the bacteria-oriented iron-siderophore complexes jointly [65]. Hydrocyanic acid, a volatile metabolite produced by various microbes, is well known for its role against various pathogens and thus plays a vital role in the biological control of phytopathogenesis [66]. Bacterial endophytes capable of producing hydrocyanic acid have been widely reported by many researchers [67].

Kaspar et al., while carrying out the isolation of endophytic bacterial isolates from *Amaranthus hybridus*, *Solanum lycopersicum*, and *Cucurbita maxima*, evaluated them for hydrocyanic acid production and observed similar findings as in the present investigation [56]. Ammonia can be secreted by microbes through numerous routes such as through nitrite ammonification, metabolism of some amino acids, and production of biogenic amines, and can be secreted also ammonia through amino acid decarboxylation deamination and urea degradation (hydrolytic) mediated by urease [68]. Most of these processes occur among soil microflora but, similar to our findings, many researchers have found endophytic bacteria such as *Bacillus* sp., *Pseudomomas* sp., and *Klebsiella* sp. growing endophytically and supplying ammonia to plant hosts so as to meet its nitrogen requirements [69]. The utility of the PCA concerns a reduction in the number of observations and concerns converting them into factors that contribute to variation; thus, in order to see the homogeneity among the isolates in this context to qualitatively study traits, the PCA helped us to determine the clustering of taxonomic groups, thus allowing us to draw relationships between the isolated microbiota in the context of traits importance. The PCA suggested a high level of biodiversity and complexity within the isolated microflora in the context of the N-fixation, ammonia secretion, HCN production, and siderophore production. Thus, this provides insight regarding the fact that most of the fungistatic strains were negative to these qualitative traits, therefore the possible mode of action in the fungistatic microflora could be the combination of them together. The PCA on all these traits also explained the niche-based role of the endophytic microflora, as explained by many investigators earlier [70,71]

The formulation used in this investigation inhibited the scab significantly at all the tested concentrations. The disease incidence and severity as a suppression measure was significantly reduced by the formulation used at higher concentrations and in certain cases, the lower concentrations were as good as the control. In our study, the results on the disease suppression suggest that endophytic bacteria can largely be tested to act as profound tools in phytopathogen management, which is in line with the 100% of diseases’ containment with bacterial endophytic cell-free supernatants as reported by several investigators across the globe [72]. The antifungal behavior of endophytic bacteria could be attributed to the biosynthesis of numerous allelo-chemicals [73]. The impact on the disease severity and disease incidence at all the concentrations could be the result of the presence of some antimicrobial compounds such as phenazines, ammonia, pyrrolnitrin, hydrocyanic acid, and pyoluterorin [74]. In *Pseudomonas* sp. and similarly in *B. subtilis* strain E1R-j, endophytic microflora from wheat (roots) revealed high fungistatic activity and thereby reduced the disease intensity (90.97%) under pot house conditions [75]. The endophytic microfloral formulations have been reported to exhibit higher blight disease containment. The mechanism employed by biological control tools to antagonize and check both the phytopathogen proliferation and advancement can be separated into four classes, namely the induction of plant defense through SAR (systemic-acquired resistance), competition, antibiosis, and finally parasitism [73]. Antibiosis largely refers to antimicrobial compounds and a huge number of antimicrobial compounds such as 2,4-diacetylphloroglucinol (DAPG), oomycin A, phenazine, tropolone, pyoluterorin, hydrogen cyanide, pyrrolnitrin, tensin, amphisinand, and cyclic lipopeptides were produced by pseudomonads [64], while xanthobaccin, oligomycin A, zwittermicin A, and kanosamine were produced by *Streptomyces* sp., *Stenotrophomonas* sp., and *Bacillus* sp. [74]. The wilt prevalence in eggplant to an extent of more than 70% was reported to be contained by the formulations of *Pseudomonas* sp., *Enterobacter* sp., and *Bacillus* sp., and it was found that most of these isolates produced an antibiotic DAPG under in vitro conditions [75]. The naturally-occurring bacterial endophytes that could confer a prominent disease resistance to plants are very low in number and such populations have to be raised to a specific level in order to encourage a contrasting check in the proliferation of phytopathogens [76].

In the present investigation, the cultivar Red Delicious revealed different levels of enzyme production, chlorophyll, and sugars upon inoculation with the bacterial formulation against the non-inoculated control. The peroxidase, superoxide dismutase, catalase, acid phosphatase, ascorbic acid peroxidase, and esterase activity varied significantly in the inoculated plants against the control. Similarly, glucose, fructose, and chlorophyll a and b content was observed to change drastically upon inoculations against the control. The present investigation is well validated with a preceding report in which microorganisms such as endophytes induce production dynamism across plant metabolites that can ameliorate the tolerance levels to various stresses [76]. In some studies, microorganisms have been found to mediate the interaction between host plants and pathogens by producing antimicrobial compounds and antioxidants, or by inducing systemic resistance [77]. Our findings are supported by the view that endophytes induce the deposition of phenolic compounds and the expression of defense-related enzymes such as peroxidases, catalases, etc. Similar to our findings [17], it was observed that joint formulation of endophytic microflora *B. amyloliquefaciens* and *P. fluorescens* amplified plant defense-based enzymes and antioxidants in formulation-treated plants, in contrast to the non-inoculated control. The discrete relationship between the formulation concentration and biochemical response, which was not identified by correlation analysis, was quantified by applying the principal component analysis. The principal component scores on PC1 were moving in the same direction between the disease severity and formulation concentrations, up to the formulation concentration magnitude of 3%, and then changed to opposite signs beyond this concentration, depicting the prominent effect of formulation concentration beyond this level. The overlapping scores of the last two concentrations revealed the peak effect values of formulation, thus giving an indication to optimize the formulation of these overlapping concentrations by maintaining narrow concentration levels in the experimental setup. Concentrations of the formulations have a significant effect on the diseases and therefore need to be optimized cautiously [51]. The colonizing behavior and biochemical response in hosts largely depend upon the formulation concentration [78].

## 5. Conclusions

This investigation confirmed the existence of potential fungistatic bacterial microflora distributed across the plethora of niches in apple germplasm. The isolated potential candidates in the form of bacterial formulation have proved to contain both disease proliferation and damage in all the investigated cultivars of apple, and this could be attributed to the modulated biochemical response in mitigating the stress mediated by the advanced infections of *V. inaequalis* in the host. This investigation provided scientific guidance for the remedies of apple scab through endophyte-mediated biological means. The formulation developed through this investigation needs to be standardized for field-level applications with respect to carrier materials, concentrations, methods of application, environmental dependences, etc., so as to enhance its efficacy. The multi-locational field experiments need to be organized to understand the behavior of these biological entities in different agro–climatic conditions. The future endeavors of this research field should be focused on aspects of transcending or minimizing the molecular and biochemical barriers that are imminent in host-specific plant–microbe interactions during biological disease management execution strategies. For the future prospects, we suggest that the substantial shrinking of the apple scab can be achieved by applying the use of these isolated strains, apart from its uses in the amelioration of plant health and development, under natural environmental conditions. Studies on the identification of their mode of action need to be carried out so as to identify some important key molecules and accordingly their production process may be amplified during their interaction as bioinoculants. The formulation needs to be tested for different application methods and optimized for better efficacy in all possible combinations so as to establish it as a more result-oriented remedy at the field level.

## Figures and Tables

**Figure 1 jof-07-00923-f001:**
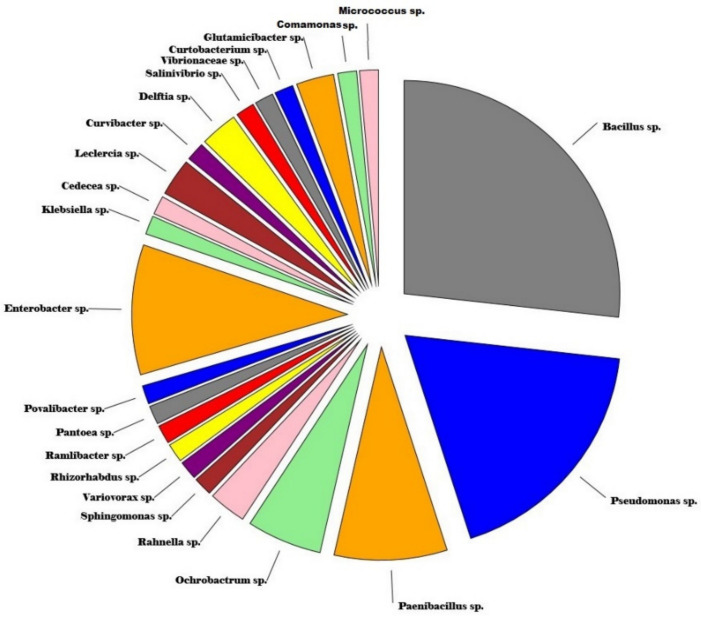
Distribution of the endophytic bacterial microflora distributed across apple germplasm (plant) having fungistatic attributes against the *Venturia inaequalis*. The genera are enumerated upon the ARDRA analysis of the 16S rRNA coding sequences of the DNA.

**Figure 2 jof-07-00923-f002:**
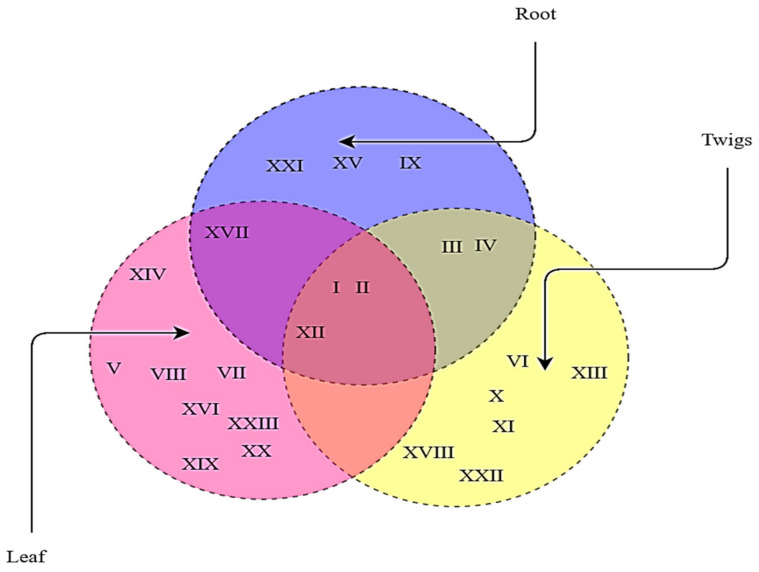
Venn diagram representing the various fungistatic microflora among the sampling plant parts. symbols in the Venn diagram represent the following: I, *Bacillus* sp.; II, *Pseudomonas* sp.; III, *Paenibacillus* sp.; IV, *Ochrobactrum* sp.; V, *Rahnella* sp.; Vi, *Sphingomonas* sp.; VII, *Variovorax* sp.; VIII, *Rhizorhabdus* sp.; IX, *Ramlibacter* sp.; X, *Pantoea* sp.; XI, *Povalibacter* sp.; XII, *Enterobacter* sp.; XIII, *Klebsiella* sp.; XIIV, *Cedecea* sp.; XV, *Leclercia* sp.; XVI, *Curvibacter* sp.; XVII, *Delftia* sp.; XVIII, *Salinivibrio* sp.; XIX, *Vibrionaceae* sp.; XX, *Curtobacterium* sp.; XXI, *Glutamicibacter* sp.; XXII, *Comamonas* sp.; and XXIII, *Micrococcus* sp.

**Figure 3 jof-07-00923-f003:**
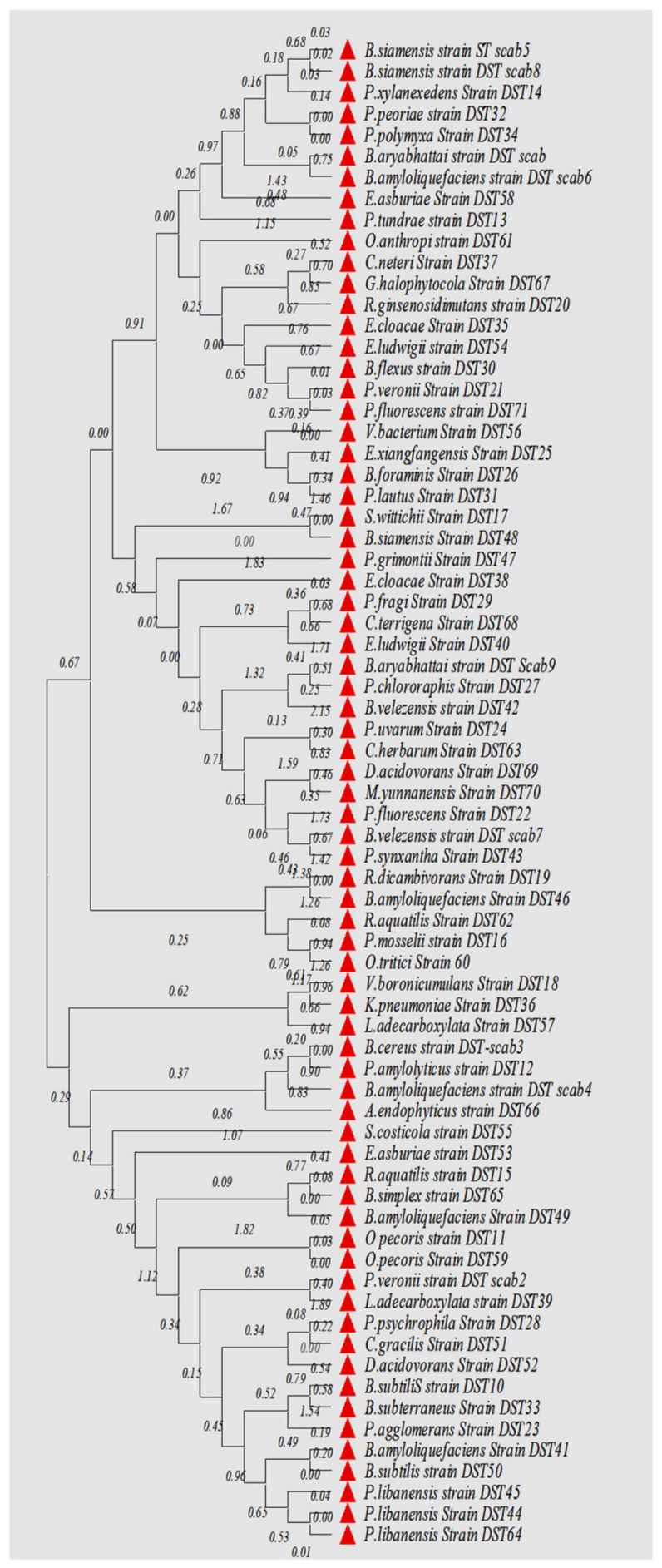
Phylogenetic tree based on the Neighbor Joining method of 16S rDNA gene sequences of isolated microflora (endophytic) with fungistatic behavior isolated from various apple plant parts. The bootstrap consensus tree anecdotal from 1000 replicates is reserved and branches corresponding to partitions duplicated in less than 50% of the bootstrap replicates are collapsed. The evolutionary distances were computed using the p-distance method and are in the units of the number of base differences per site. The optimal tree with the sum of the branch length = 20.71250806 is shown. The confidence probability that the interior branch length is greater than 0 is estimated using the bootstrap test (shown next to the branches). This analysis involved 71 nucleotide sequences. Codon positions included were 1st + 2nd + 3rd + no-coding. All positions with less than 95% site coverage were eliminated, i.e., fewer than 5% of alignment gaps, missing data, and ambiguous bases were allowed at any position (partial deletion option). There were a total of 1389 positions in the final dataset. Evolutionary analysis was conducted in MEGA X (v. 10.2.6).

**Figure 4 jof-07-00923-f004:**
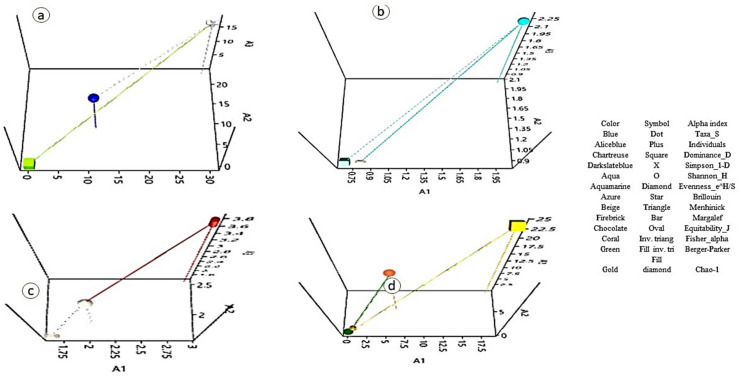
Alpha diversity indices ((**a**) Taxa S, individuals, and dominance; (**b**) Simpson, Shanon, and Evenness; (**c**) Brillouin, Menhinick, and Margalef; and (**d**) equitability, Fisher alpha, Berger Parker, and Chao-1) of the fungistatic microflora isolated from various plant parts across the apple germplasm. The diversity indices reflected a higher diversity in twigs (**A1**), followed by roots (**A2**) and leaves (**A3**) with respect to the presence of bacterial microflora fungistatic to *Venturia inaequalis*, as represented in the diagram.

**Figure 5 jof-07-00923-f005:**
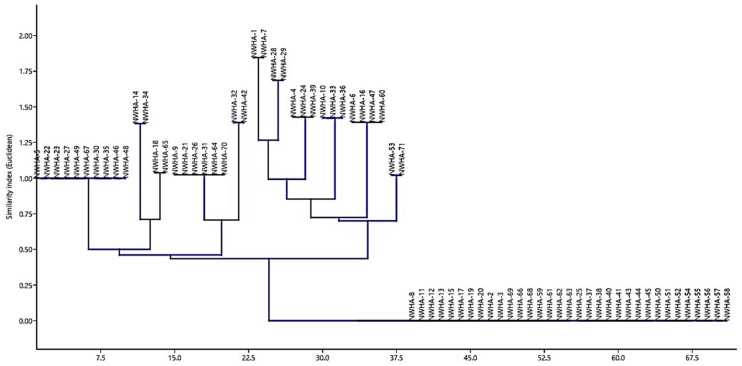
Cluster diagram based on the similarity index (Euclidian) in the context of the various functional traits (NH_3_ production, N_2_ fixation, siderophore production, and HCN production) pertaining to isolated bacterial microfloral endosymbiosiomes of apple. The microflora with homogeneity in the functional attributes were clubbed, as shown in the diagram.

**Figure 6 jof-07-00923-f006:**
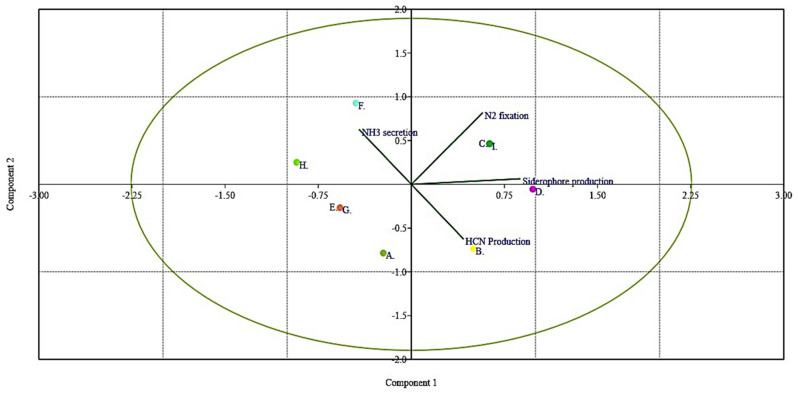
Principal component analysis (PCA) based on the qualitative assessment of NH_3_ production, N_2_ fixation, siderophore production, and HCN production (represented by the navigating lines) pertaining to isolated fungistatic bacterial microflora against *Venturia inqualis* of the apple germplasm. The PCA of the qualitative metabolite production revealed that principal component 1 (PC1) and principal component 2 (PC2) accounted for 38.733% and 29.572% of the data variation, respectively. Letters in the figure represent the different clusters generated in the cluster analysis.

**Figure 7 jof-07-00923-f007:**
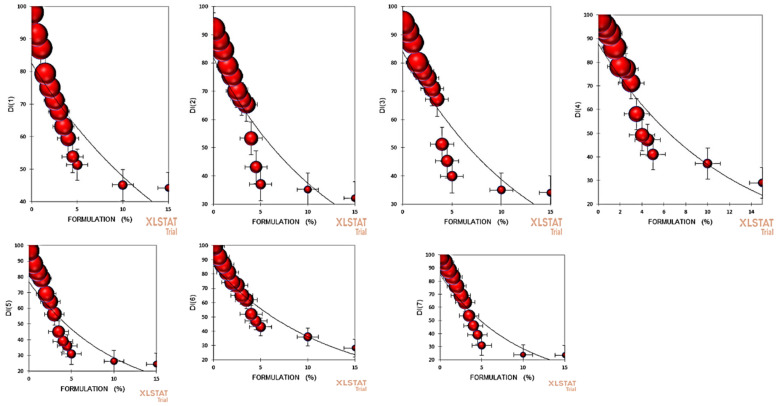
In the scatter plot, **DI** (**1**) represents the disease incidence in Red Delicious; **DI** (**2**) represents the disease incidence in Gala Redlum; **DI** (**3**) represents the disease incidence in Early Red One; **DI** (**4**) represents the disease incidence in Red Fuji; **DI** (**5**) represents the disease incidence in Gala Mast; **DI** (**6**) represents the disease incidence in Red Chief; and **DI** (**7**) represents the disease incidence in Summer Red, all vs. formulation percentage. Balls of different size indicate the disease intensity.

**Figure 8 jof-07-00923-f008:**
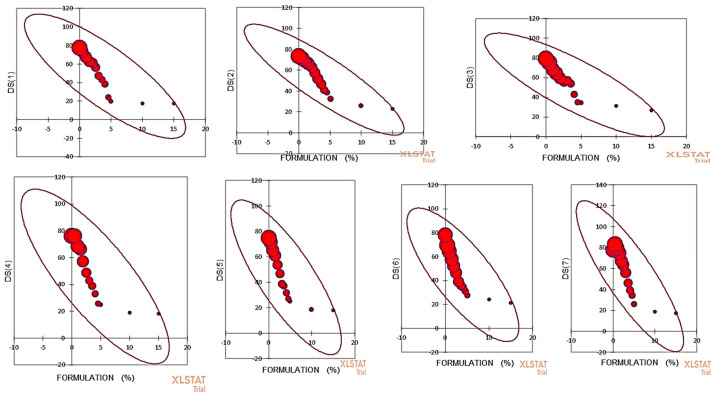
In the scatter plot, **DS** (**1**) represents the disease severity in Red Delicious; **DS** (**2**) represents the disease severity in Gala Redlum; **DS** (**3**) represents the disease severity in Early Red One; **DS** (**4**) represents the disease severity in Red Fuji; **DS** (**5**) represents the disease severity in Gala Mast; **DS** (**6**) represents the disease severity in Red Chief; and **DS** (**7**) represents the disease severity in Summer Red, all vs. formulation percentage. Balls of different size indicate the disease severity.

**Figure 9 jof-07-00923-f009:**
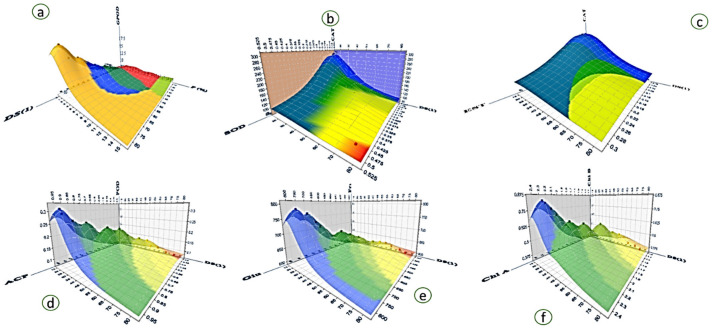
Plot representing potent endophyte formulation impact on the Red Delicious variety of apple: (**a**) disease severity and GPOD activity as influenced by formulation percentage; (**b**) effect on CAT and SOD; (**c**) effect on CAT and EST; (**d**) effect on POD and ACP; (**e**) effect on glucose and fructose; and (**f**) effect on Chlorophyll a and Chlorophyll b. The biochemical response was recorded in the apple leaves. The abbreviations indicate: peroxidase (GPOD), catalase (CAT), superoxide dismutase (SOD), ascorbic acid peroxidase (POD), esterase (EST), and acid phosphatase (ACP).

**Figure 10 jof-07-00923-f010:**
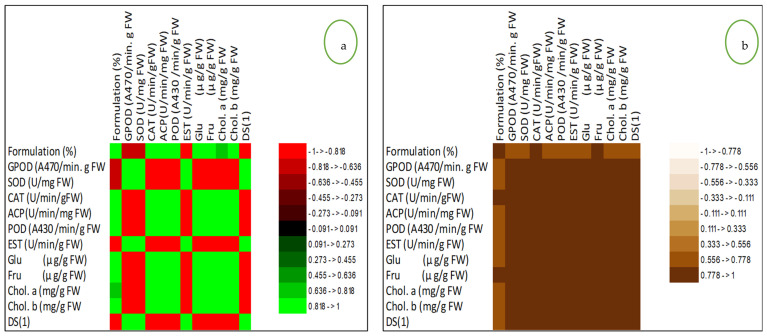
(**a**): Correlation matrix (Pearson) and (**b**): Coefficients of determination (Pearson) of disease severity (DS1), formulation percentage and physiological attributes. Abbrevations indicate: peroxidase (GPOD), catalase (CAT), superoxide dismutase (SOD), ascorbic acid peroxidase (POD), esterase (EST), acid phosphatase (ACP)] in Red Delicious apple variety. The biochemical response was recorded in the apple leaves.

**Figure 11 jof-07-00923-f011:**
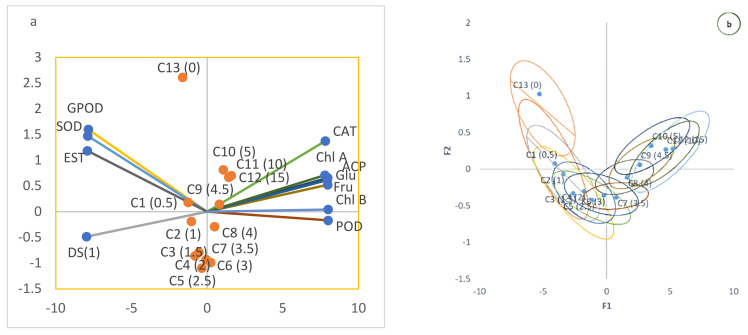
Principal component analysis (PCA) of disease severity (DS1) and various defense enzymes; peroxidase (GPOD), catalase (CAT), superoxide dismutase (SOD), ascorbic acid peroxidase (POD), esterase (EST), and acid phosphatase (ACP). Disease severity (DS1), chlorophyll, glucose, and fructose with respect to different formulation concentrations are represented by solid colored dots in A and C in b. The PCA of the defense response to bacterial formulation in Red Delicious revealed that principal component 1 (**F1**) and principal component 2 (**F2**) accounted for 97.28% and 1.739% of the data variance, respectively. The first graph (**a**) represents the biplot of the axes in which different variables are represented by different colored axes and the second graph (**b**) represents the bootstrap ellipses in which concentrations of the formulation are represented by C and their corresponding value is given in brackets.

**Table 1 jof-07-00923-t001:** Magnitude of antifungal behavior and plant growth-promoting attributes of the isolated microflora.

Strain	PGI (%)	IAA (µg/mL)	Phosphate Solubilization		ACC Deaminase (µM mg^−1^ h^−1^)
			QS (µg/mL)	S. I.	
DST scab	81.95 ± 0.03 ^(F)^	146.05 ± 0.53 ^(a)^	0.00	0.00	0.00
DST scab2	54.75 ± 0.12 ^(X)^	8.94 ± 0.10 ^(y)^	117.24 ± 0.05 ^(k)^	1.67 ± 0.07 ^(e)^	0.00
DST scab3	45.6 ± 0.01 ^(A6)^	93.13 ± 0.18 ^(g)^	0.00	0.00	0.00
DST scab4	82.55 ± 0.07 ^(E)^	0.00	143.50 ± 0.02 ^(i)^	2.30 ± 0.22 ^(b)^	19.31 ± 0.72 ^(b)^
DST scab5	67.4 ± 0.18 ^(O)^	11.75 ± 0.23 ^(w)^	0.00	0.00	0.00
DST scab6	45.20 ± 0.09 ^(A6)^	0.00	0.00	0.00	8.07 ± 0.12 ^(g)^
DST scab7	78.60 ± 0.61 ^(H)^	0.00	0.00	0.00	0.00
DST scab8	66.2 ± 0.83 ^(P)^	28.92 ± 0.10 ^(t)^	0.00	0.00	0.00
DST Scab9	67.45 ± 0.18 ^(O)^	31.76 ± 0.83 ^(r)^	0.00	0.00	13.31 ± 0.136 ^(e)^
DST10	50.10 ± 0.08 ^(A2)^	0.00	0.00	0.00	0.00
DST11	24.35 ± 0.45 ^(h)^	0.00	125.08 ± 0.58 ^(j)^	2.20 ± 0.09 ^(b)^	0.00
DST12	65.25 ± 0.05 ^(Q)^	0.00	0.00	0.00	0.00
DST13	63.05 ± 0.19 ^(R)^	49.03 ± 0.63 ^(n)^	0.00	0.00	0.00
DST14	77.55 ± 0.92 ^(I)^	0.00	218.04 ± 0.06 ^(b)^	2.04. ± 0.53 ^(c)^	0.00
DST15	85.7 ± 0.28 ^(C)^	15.82 ± 0.13 ^(v)^	0.00	0.00	14.42 ± 0.06 ^(d)^
DST16	58.01 ± 0.13 ^(U)^	71.15 ± 0.59 ^(l)^	0.00	0.00	0.00
DST17	66.07 ± 0.01 ^(P)^	0.00	0.00	0.00	0.00
DST18	58.40 ± 0.07 ^(U)^	1.16 ± 0.23 ^(a1)^	156.43 ± 0.23 ^(g)^	2.18 ± 0.03 ^(b)^	0.00
DST19	32.10 ± 0.017 ^(f)^	44.18 ± 0.39 ^(p)^	0.00	0.00	0.00
DST20	45.35 ± 0.10 ^(A6)^	0.00	0.00	0.00	0.00
DST21	54.75 ± 0.01 ^(y)^	29.98 ± 0.17 ^(s)^	188.82 ± 0.14 ^(e)^	2.52 ± 0.09 ^(a)^	0.00
DST22	48.15 ± 0.79 ^(A3)^	41.29 ± 0.05 ^(q)^	225.51 ± 0.01 ^(a)^	1.93 ± 0.15 ^(d)^	0.00
DST23	15.01 ± 0.04 ^(k)^	0.00	0.00	0.00	0.00
DST24	71.85 ± 0.13 ^(M)^	0.00	0.00	0.00	6.41 ± 1.02 ^(h)^
DST25	23.80 ± 0.43 ^(i)^	0.00	0.00	0.00	0.00
DST26	38.04 ± 0.23 ^(b)^	77.52 ± 0.04 ^(j)^	191.04 ± 0.15 ^(d)^	2.15 ± 0.19 ^(b)^	0.00
DST27	52.01 ± 0.02 ^(A1)^	0.00	0.00	0.00	0.00
DST28	96.35 ± 0.03 ^(A)^	48.25 ± 0.19 ^(o)^	0.00	0.00	18.21 ± 0.00 ^(c)^
DST29	91.25 ± 0.11 ^(B)^	31.65 ± 0.83 ^(r)^	151.13 ± 0.09 ^(h)^	2.53 ± 0.01 ^(a)^	0.00
DST30	50.25 ± 0.40 ^(A2)^	0.00	0.00	0.00	0.00
DST31	13.05 ± 0.19 ^(l)^	101.29 ± 0.87 ^(f)^	0.00	0.00	0.00
DST32	75.19 ± 0.00 ^(K)^	87.05 ± 0.65 ^(h)^	162.54 ± 0.03 ^(f)^	2.52 ± 0.19 ^(a)^	0.00
DST33	83.05 ± 0.42 ^(D)^	0.00	0.00	0.00	0.00
DST34	54.25 ± 0.02 ^(X)^	71.96 ± 0.69 ^(k)^	0.00	0.00	0.00
DST35	43.75 ± 0.21^(a)^	0.00	0.00	0.00	0.00
DST36	80.85 ± 0.49 ^(G)^	51.09 ± 0.12 ^(m)^	124.23 ± 0.09 ^(j)^	1.85 ± 0.07 ^(d)^	0.00
DST37	66.25 ± 0.00 ^(P)^	0.00	0.00	0.00	0.00
DST338	46.06 ± 0.36 ^(A5)^	81.65 ± 0.03 ^(i)^	0.00	0.00	0.00
DST39	77.15 ± 1.91 ^(I)^	0.00	0.00	0.00	0.00
DST40	68.08 ± 0.17 ^(N)^	0.00	0.00	0.00	0.00
DST41	67.34 ± 0.22 ^(O)^	123.07 ± 0.14 ^(b)^	0.00	0.00	0.00
DST42	47.75 ± 0.02 ^(A4)^	21.08 ± 0.00 ^(u)^	0.00	0.00	0.00
DST43	23.75 ± 0.09 ^(i)^	0.00	0.00	0.00	0.00
DST44	68.65 ± 0.14 ^(N)^		0.00	0.00	0.00
DST45	52.75 ± 0.32 ^(Z)^	0.00	203.08 ± 0.02 ^(c)^	1.93 ± 0.06 ^(d)^	0.00
DST46	76.25 ± 1.11 ^(J)^	11.16 ± 0.27 ^(x)^	0.00	0.00	0.00
DST47	68.48 ± 0.06 ^(N)^	117.17 ± 1.30 ^(c)^	0.00	0.00	0.00
DST48	57.75 ± 0.88 ^(V)^	0.00	0.00	0.00	0.00
DST49	67.75 ± 0.02 ^(O)^	0.00	0.00	0.00	0.00
DST50	57.04 ± 0.11 ^(V)^	0.00	0.00	0.00	22.61 ± 0.72 ^(a)^
DST51	37.45 ± 0.09 ^(c)^	0.00	0.00	0.00	0.00
DST52	45.16 ± 0.52 ^(A6)^	2.25 ± 0.06 ^(z)^	0.00	0.00	0.00
DST53	55.75 ± 0.28 ^(W)^	0.00	0.00	0.00	0.00
DST54	45.25 ± 0.66 ^(A6)^	0.00	0.00	0.00	0.00
DST55	47.15 ± 0.06 ^(A4)^	110.05 ± 1.13 ^(e)^	0.00	0.00	0.00
DST56	67.25 ± 0.26 ^(O)^	1.19 ± 0.01 ^(a1)^	0.00	0.00	0.00
DST57	18.95 ± 0.07 ^(j)^	81.54 ± 0.73 ^(i)^	0.00	0.00	0.00
DST58	36.13 ± 0.08 ^(d)^	0.00	0.00	0.00	0.00
DST59	11.25 ± 0.08 ^(l)^	0.00	0.00	0.00	0.00
DST60	73.5 ± 0.09 ^(L)^	0.00	0.00	0.00	11.91 ± 0.36 ^(f)^
DST61	59.17 ± 0.02 ^(T)^	71.59 ± 1.12 ^(kl)^	102.16 ± 0.05 ^(m)^	2.19 ± 0.07 ^(b)^	0.00
DST62	46.15 ± 0.05 ^(A5)^	0.00	0.00	0.00	0.00
DST63	18.25 ± 0.16 ^(j)^	0.00	0.00	0.00	0.00
DST64	58.05 ± 0.37 ^(U)^	0.00	115.65 ± 0.01 ^(l)^	1.34 ± 0.18 ^(f)^	0.00
DST65	38.25 ± 0.14 ^(b)^	0.00	0.00	0.00	0.00
DST66	30.25 ± 0.33 ^(g)^	0.00	0.00	0.00	0.00
ST67	62.22 ± 0.18 ^(S)^	115.75 ± 0.93 ^(d)^	0.00	0.00	0.00
DST68	53.75 ± 0.02 ^(Y)^	0.00	0.00	0.00	0.00
DST69	33.75 ± 0.16 ^(e)^	8.75 ± 0.05 ^(y)^	0.00	0.00	0.00
DST70	63.19 ± 0.07 ^(R)^	0.00	0.00	0.00	7.94 ± 0.26 ^(g)^
DST71	47.25 ± 0.81 ^(A4)^	0.00	0.00	0.00	0.00

Subscripts indicate that within the column the values with same letters did not differ by Tukey’s test.

**Table 2 jof-07-00923-t002:** Output for the PCA of the qualitative plant growth-promoting attributes pertaining to isolated endophytic bacterial microflora.

Bacterial Strains	CL	N	AN	CN	S
NWHA-18, NWHA-65	A	0	0	1	0
NWHA-4, NWHA-24, NWHA-39	B	0	0	1	1
NWHA-7, NWHA-1	C	1	1	1	1
NWHA-28, NWHA-29	D	1	0	1	1
NWHA-8, NWHA-2, NWHA-3, NWHA-11, NWHA-12, NWHA-13, NWHA-15, NWHA-17, NWHA-19, NWHA-20, NWHA-25, NWHA-37, NWHA-38, NWHA-40, NWHA-41, NWHA-43, NWHA-44, NWHA-45, NWHA-50, NWHA-51, NWHA-52, NWHA-54, NWHA-55, NWHA-56, NWHA-57, NWHA-58, NWHA-59, NWHA-61, NWHA-62, NWHA-63, NWHA-66, NWHA-68, NWHA-69	E	0	0	0	0
NWHA-32, NWHA-42	F	1	1	0	0
NWHA-14, NWHA-34	G	0	1	1	0
NWHA-30, NWHA-5, NWHA-22, NWHA-23, NWHA-27, NWHA-35, NWHA-46, NWHA-48, NWHA-49, NWHA-67	H	0	1	0	0
NWHA-10, NWHA-33, NWHA-36	I	1	0	0	1
NWHA-9, NWHA-21, NWHA-26, NWHA-31, NWHA-64, NWHA-70,	J	1	0	0	0
NWHA-53, NWHA-71,	K	0	0	0	1
NWHA-6, NWHA-16, NWHA-47, NWHA-60	L	0	1	0	1

CL: Cluster; N: N-fixation; AN: Ammonia secretion; CN: HCN production and S: siderophore production.

**Table 3 jof-07-00923-t003:** Biochemical and cultural features of formulations forming endophytic microflora.

Attribute	a	b	c	d	e	f	g	h	i	j
1.	-	-	-	-	-	-	-	-	-	-
2.	+	+	+	-	-	+	-	-	-	
3.	Rods	Rods	Rods	Rods	Rods	Rods	Rods	Rods	Rods	Rods
4.	C	C	C	-	-	-	-	-	-	T
5.	-	-	-	-	+	-	+	+	+	-
6.	-	-	-	-	+	-	-	-	+	-
7.	-	-	-	-	+	-	+	-	-	-
8.	+	+	+	+	-	+	+	-	-	+
9.		+	+	+	+	+	-	+	-	+
10.	+	+	+	+	+	+	+	+	+	+
11.	+	+	+	-	-	-	-	-	-	+
12.	+	+	+	+	+	-	+	+	+	-
13.	-	+	+	-	-	+	-	-	-	-
14.	+	-	-	-	+	+	+	-	+	+
15.	+	-	-	-	+	+	+	+	+	+
16.	+	+	+	+	+	+	+	+	+	+
17.	-	+	+	-	+	+	+	+	+	-
18.	-	+	+	-	+	+	+	+	+	+
19.	+	+	+	-	-	+	-	-	-	+
20.	+	-	-	-	-	+	-	-	-	+
21.	+	-	-	-	+	+	+	-	+	+
22.	+	+	+	-	-	+	-	-	-	-
23.	-	-	-	-	+	-	+	-	+	-
24.	-	+	-	-	-	-	-	-	-	-
25.	+	+	+	+	+	+	+	+	+	+
26.	-	+	-	-	-	+	-	-	-	+
27.	-	-	+	-	+	-	+	-	+	+
28.	-	-	-	+	+	-	+	+	+	-
29.	+	+	-	-	-	+	-	-	-	-
30.	+	-	+	+	-	+	-	+	-	+
31.	-	-	-	-	-	+	-	-	-	-

a: Bacillus aryabhattai strain DST scab; b: Bacillus amyloliquefaciens strain DST scab4; c: Bacillus velezensis strain DST scab7; d: Pseudomonas fragi strain DST29; e: Rahnella aquatilis strain DST15; f: Paenibacillus xylanexedens strain DST14; g: Klebsiella pneumoniae strain DST36; h: Pseudomonas psychrophila strain DST28; i: Leclercia adecarboxylata strain DST39; j: Bacillus subterraneus strain DST33; 1: pigment production; 2: gram reaction; 3: shape; 4: endospore position; 5: indole production; 6: methyl red test; 7: Voges–Proskauer reaction; 8: citrate utilization; 9: oxidase; 10: catalase; 11: H_2_S production; 12: starch hydrolysis; 13: cellulose hydrolysis; 14: acid production; 15: glucose; 16: sucrose; 17: lactose; 18: maltose; 19: amylase; 20: gelatinase; 21: L-arabinose; 22: L-xylose; 23: D-xylose; 24: glutamate decarboxylase; 25: tryptophan deaminase; 26: casein hydrolysis; 27: D-sorbitol; 28: nitrate reduction; 29: lysine decarboxylase; 30: gelatin hydrolysis; and 31: methyl-beta-D-xylopyranoside. C = central and T = terminal.

**Table 4 jof-07-00923-t004:** Molecular aspects of isolated microflora based on data retrieved from the National Center for Biotechnology Information, USA.

Strain	Distance Tree Results			Graphics	
	Similarity (%)	E Value	Acc. Len	Max. Score	Total Score
DST scab	99.61	0.00	1533	2832	2832
DST scab4	100.0	0.00	1553	2868	2868
DST scab7	99.00	0.00	1560	2881	2881
DST scab14	99.14	0.00	1516	2819	2819
DST scab15	99.13	0.00	1505	2780	2780
DST scab28	99.15	0.00	1537	2839	2839
DST scab29	99.52	0.00	1462	2693	2693
DST scab33	99.87	0.00	1522	2811	2811
DST scab36	99.87	0.00	1498	2767	2767
DST scab39	9.93	0.00	1449	2676	2676

## Data Availability

Data sharing not applicable to this article.

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
