# Peer review of "Bacterial Endophyte Community Dynamics in Apple (Malus domestica Borkh.) Germplasm and Their Evaluation for Scab Management Strategies"

_jof, 2021, doi:10.3390/jof7110923_

Round 1

Reviewer 1 Report

This study is aimed to investigate a biological approach in which bacterial endophyte community dynamics was examined across the apple germplasm in relation to the fungistatic activity against Venturia inaequalis.The study shows an acceptable design and some original aspects. There are some interesting points of the results that can be published in Journal fo Fungi. However, this study may be suitable only after careful revisions.

Comments and suggestions:

V. inaequalis instead of Venturia inaequalis in L17, L25

L28: you should put in double quote ’Red Delicious’

A conclusion is missing at the end of the Abstract

L34: Malus domestica - in italic

L41: Venturia inequalis, Podosphaera leucotrichia, Alternaria mali - in italic

L42-44: V. inequalis – in italic

L64, L83, L87: V. inaequalis instead of Venturia inaequalis. This should be done throughout the manuscript.

L85-88: This sentence is too compliacted. Please reword.

L98-L99 and L105, L111: V. inequalis – in italic

L101: C means celsius? Then you need to correct ’C with superscript o’ and not with 0.

L122: underscipt problems of numbers with chemicals such as Na2HPO4. This should be done throughout the text

L125: Do you mean 37 celsius and not 370 celsius???

L246: in vivo – in italic

L347-362: All latin names e.g. Pseudomonas fluorescens and Bacillus subtilis Bacillus amyloliquefaciens Povalibacter uvarum – should be in italic. This also should be done throughout the text.

L486: Venturia inaequalis – in italic

L667: Give more strong conclusions. Give your conclusions in points.

References are not follow journal format. E.g. All authors name should be given. Check carefully the preparation guideline

One of my suggestion for further reading is: Köhl, J et al. "Toward an integrated use of biological control by Cladosporium cladosporioides H39 in apple scab (Venturia inaequalis) management." Plant Disease 99.4 (2015): 535-543.

Author Response

We appreciate your valuable suggestions and comments. All your valuable suggestions were taken into consideration and changes have been incorporated in revised MS

Reviewer 2 Report

In this study the biocontrol ability of bacterial endophytes isolated from the leaves, twigs and roots towards Apple scab were investigated. The authors further characterised the microbial community, and selected individuals with increased potency agains apple scab to prepare formulations. These strains were further characterised for their PGP abilities.

For a detailed report, please see the attached PDF.

The manuscript requires extensive corrections of language and grammar use as indicated in the attached reviewer report. In addition, the authors switch between the full and abbreviated names of organisms. The names of organisms should also be in italics throughout the manuscript. 

Introduction:

The introduction is vague, and needs to be improved. There is not mention of the currently used biocontrol methods against V. inaequalis, or any previous studies using endophytes to control apple scab (this is briefly mentioned in the discussion). It is also  not clear how this research will fill the current gap in the literature.

Materials and methods:

This section is too vague. The authors should use the metric system for all their values, and not switch between metric and imperial. There are a lot of mistakes in this section as indicated in the attached PDF document. 

It is not clear how old the plants used in this study were, and which leaves (number) were used for endophyte isolation and scab infection. 

In section 2.7, it is not clear how the nucleotide sequences were obtained that the authors used in the alignments. Was it sequenced? What platform was used?

Section 2.8.4 and 2.8.5 should be combined, and linked together. 

The authors should clearly mention the different apple varieties and their genetic makeup in the materials and methods.

It is not clear what is represented in Figure 1. Did the authors pool the samples from the roots, twigs and leaves to obtain a total representation? If not, it will be more interesting to show the distribution for each tissue separately, followed by a Venn diagram to show the overlap between tissue types.

The control treatment used to compare the formulations with is not explained in the materials and methods. Was this a water dipping?

The details on how the different Principle Component Analysis were performed should be explained. How did the authors scale and/or normalise the different datasets in order to combine them?

Results:

In general, the figure quality is not good enough for publication. Higher resolution images should be used. Figure 12, move A and B to the top left.

Table 1 can be moved to the supplementary data.

In figure 2, the text overlap with the lines and should be corrected.

Why was PCA analysis selected to represent the data? There is clearly no clustering of any of the data presented in PCA analysis. In this case a PCoA will be clearer. The authors also do not mention this in the results or discussion. The distribution of the different datasets on the PCA should be explained, and not merely referring to which component explains most of the phenotypes. 

I suggest to replace Figure 4 with a Venn diagram showing the different distribution of organisms between the three tissue types, and which organisms overlap. The current figure is too complicated and no clear correlation can be made.

Figure 6 again is difficult to interpret. I suggest to simply show this in a table.

Figure 7 is also too complicated, and it is not explained how the tree was constructed. 

In figure 8 and 9, what does the size of the red dots represent? Mention this in the figure legends.

The data represented in Figure 10, 11 and 12 should be explained in the results. The authors overcomplicated these figures with no justification as to why these methods were selected. For a PCA analysis, the authors should explain that it was performed to look for correlation between the different datasets. The authors should then dissect the output of the PCA to make it clear for the reader what is the message hidden in the graph. This is very vague in the results section, and not discussed at al in the discussion. 

Discussion:

The discussion lack structure. The purpose of this study should be made clear in the first paragraph. The authors merely mention references to support their findings, but the results of the study is not discussed. 

Which microbes showed the highest protection and why? How does this correlate with all the compounds and enzyme activity measured studied? How does the distribution of the different microorganisms across the different apple varieties correlate with their native immunity against apple scab? Are these formulations something that can be applied in the field? These are the key things to discuss.

Conclusion:

How can this study be used in the field? What are the next steps, and what is still outstanding from this study that should be considered in follow up studies?

Author Response

(The authors gave the same response as above.)

Reviewer 3 Report

Review of Bacterial Endophyte Community Dynamics in Apple (Malus domestica borkh) and their Evaluation and Management

Scab and related fungi cause major loses in fruit.  This fairly long and interesting report describes antifungal behavior in 155 bacterial isolates from 19 Apple Cultivars.  A Strength of this study is that it describes biochemical properties of these bacteria in detail.  I think this will be a useful addition to the literature.   I have some suggestions which perhaps may be useful.

APPLE LOSES DUE TO SCAB.  Are there any papers which estimated approximate losses of apples and other fruit to scab.  Perhaps adding a reference or two and a sentence or two to describe apple scab losses might be interesting.

POSSIBLE HUMAN PATHOGENS.  I note that at least two of bacteria isolated are potential human pathogens- Klebsiella pneumonia and Enterobacter cloacae.  Do these cause any human health concerns?

ANY IDEAS TO USE BACTERIAL ENDOPHYTES AS TREATMENT?  In the Kashmir region do they use bacterial treatments such as  Bacillus subtilis .

What suggestions do you have for the future treatment regimes?  

Author Response

(The authors gave the same response as above.)

Round 2

Reviewer 2 Report

Dear authors,

Thank you for addressing my concerns in the new version of the manuscript. The manuscript is much improved, but I still have a few remarks. 

Firstly, there are still plenty of grammar mistakes as indicated in the attached PDF. I advise the authors to carefully go through the manuscript using the attached PDF as a guide and introduce the suggested corrections. For instance, the word "percent" is written as "per cent" throughout and should be corrected. When referring to a figure in text, the authors use a combination of "Fig., Figure and fig.", keep it consistent as "Fig. x" throughout. A lot of the sentences in the manuscript are extremely long, and contains information not related to one another. I suggest to simplify these long sentences and make them more concise. A lot of sentences also do not end with a full stop and should be amended. 

Methods:

Section 2.5, the morphological characterisation is explained in section 2.8.10, and should be moved to section 2.5.

In Figure 6, it is not clear what the letters on the PCA plot are referring to.

All figures should be placed within the body of the results section after they are first mentioned in the text. 

In Figure 7, the legend suggests that the ball size is linked to formulation%, but it is in fact linked to disease index. Please clarify this.

In Figure 9, I suggest that the authors keep the disease index (DI) and disease severity always on the same axis between the different figures. It is easier to see how the other parameters influences DI or DS when these are kept on the same axis.

Discussion:

There is still a missing link in the discussion between the correlation between the presence of specific endophytes and the apple varieties natural resistance. Can you make any such correlation from your data? Does the variety with the highest natural resistance against Venturia contain the highest amount of antagonistic endophytes? 

There should also be a discussion on why these plants do not show natural resistance against Venturia even though they naturally contain the endophytic bacteria used in this study. Are the bacterial load too low, or do you require a combination of these bacteria for effective protection?

There is one extremely long paragraph in the discussion (more than one page), I suggest to split it into smaller paragraphs.

In the conclusion, the authors should propose how such a strategy can be implemented in the field. What will be the most effective, using a spray formulation on the leaves, or performing root inoculations before apple seedlings are planted. Will this be feasible for farmers?

Author Response

We are very much thankful to the reviewer for the encouragement positive feedback to enhance the quality of our manuscript.  Fig. Has been kept constant, Sentence has been made precise and concise, Typo and other errors also rectified

The authors would like to acknowledge the reviewer for his noble comments. Care has been taken to improve the presentation of the work and address the concerns raised as per the specific comments below.

Q: Firstly, there are still plenty of grammar mistakes as indicated in the attached PDF. I advise the authors to carefully go through the manuscript using the attached PDF as a guide and introduce the suggested corrections. For instance, the word "percent" is written as "per cent" throughout and should be corrected. When referring to a figure in text, the authors use a combination of "Fig., Figure and fig.", keep it consistent as "Fig. x" throughout. A lot of the sentences in the manuscript are extremely long, and contains information not related to one another. I suggest to simplify these long sentences and make them more concise. A lot of sentences also do not end with a full stop and should be amended. 

Response: We are very much thankful to the reviewer for the encouragement positive feedback to enhance the quality of our manuscript.  Fig. Has been kept constant, Sentence have been made precise and consise, Typo and other errors rectified

Q: Methods: Section 2.5, the morphological characterisation is explained in section 2.8.10, and should be moved to section 2.5.

Response: The characterization has been done only in the isolates which were used in making the formulation. Therefore, while adding them in the suggested section would make the method vague

Q: In Figure 6, it is not clear what the letters on the PCA plot are referring to.

Response: Principal component analysis (PCA) based on qualitative assessment of NH3 production, N2 fixation, siderophore production and HCN production (represented by the navigating lines) pertaining to isolated fungistatic bacterial microflora against Venturia inqualis of apple germplasm. PCA of the qualitative metabolite production revealed that principal component 1 (PC1) and principal component 2 (PC2) accounted for 38.733 and 29.572 % of the data variation, respectively. Letters in the figure represent the different clusters generated in the cluster analysis.

Q: All figures should be placed within the body of the results section after they are first mentioned in the text

Response: Needful done

Q: In Figure 7, the legend suggests that the ball size is linked to formulation%, but it is in fact linked to disease index. Please clarify this.

Response: Incorporations done as suggested.

Q: In Figure 9, I suggest that the authors keep the disease index (DI) and disease severity always on the same axis between the different figures. It is easier to see how the other parameters influences DI or DS when these are kept on the same axis

Response: The authors have tried all the possible combinations but the real message is reflected only in the current arrangement.

Discussion:

There is still a missing link in the discussion between the correlation between the presence of specific endophytes and the apple varieties natural resistance. Can you make any such correlation from your data? Does the variety with the highest natural resistance against Venturia contain the highest amount of antagonistic endophytes? 

Response: Needful done.

Q: There should also be a discussion on why these plants do not show natural resistance against Venturia even though they naturally contain the endophytic bacteria used in this study. Are the bacterial load too low, or do you require a combination of these bacteria for effective protection? There is one extremely long paragraph in the discussion (more than one page), I suggest to split it into smaller paragraphs.

Response: The extent of endophyte colonization in plants by endophytes has been observed to be correlated to the disease resistance. The naturally occurring bacterial endophytes that could confer a prominent disease resistance to the plants are in very low in number and such their populations have to be raised to a level to bring a contrasting check in the proliferation of phytopathogens.

The formulation needs to be tested for different application methods and optimized for better efficacy in all possible combinations, so as to make it more result-oriented remedy at field level.

Q: In the conclusion, the authors should propose how such a strategy can be implemented in the field. What will be the most effective, using a spray formulation on the leaves, or performing root inoculations before apple seedlings are planted. Will this be feasible for farmers?

Response:  The formulation needs to be tested for different application methods and optimized for better efficacy in all possible combinations, so as to make it more result-oriented remedy at field level. The formulation needs some standardization in context to best possible application method and carrier material before it could reduce the scab by a large extent.